# Efficiency of the Cerebroplacental Ratio in Identifying High-Risk Late-Term Pregnancies

**DOI:** 10.3390/medicina59091670

**Published:** 2023-09-15

**Authors:** Raquel Martin-Alonso, Valeria Rolle, Ranjit Akolekar, Catalina de Paco Matallana, Irene Fernández-Buhigas, Maria Isabel Sánchez-Camps, Tara Giacchino, Miguel Rodríguez-Fernández, Jose Eliseo Blanco-Carnero, Belén Santacruz, María M. Gil

**Affiliations:** 1Department of Obstetrics and Gynecology, Hospital Universitario de Torrejón, Torrejón de Ardoz, 28850 Madrid, Spain; 2Faculty of Medicine, Universidad Francisco de Vitoria, Pozuelo de Alarcón, 28223 Madrid, Spain; 3Statistics and Data Management Unit, iMaterna Foundation, Alcalá de Henares, 28806 Madrid, Spain; 4Facultad de Estudios Estadísticos, Universidad Complutense de Madrid, 28040 Madrid, Spain; 5Medway Fetal and Maternal Medicine Centre, Medway NHS Foundation Trust, Gillingham M75NY, UK; 6Institute of Medical Sciences, Canterbury Christ Church University, Chatham CT11QU, UK; 7Hospital Clínico Universitario Virgen de la Arrixaca, El Palmar, 30120 Murcia, Spain; 8Institute for Biomedical Research of Murcia, IMIB-Arrixaca, El Palmar, 30120 Murcia, Spain

**Keywords:** adverse perinatal outcome, middle cerebral artery Doppler, umbilical artery Doppler, cesarean section, stillbirth, fetal growth restriction, late-term

## Abstract

*Background and Objectives:* Over the last few years, great interest has arisen in the role of the cerebroplacental ratio (CPR) to identify low-risk pregnancies at higher risk of adverse pregnancy outcomes. This study aimed to assess the predictive capacity of the CPR for adverse perinatal outcomes in all uncomplicated singleton pregnancies attending an appointment at 40–42 weeks. *Materials and Methods*: This is a retrospective cohort study including all consecutive singleton pregnancies undergoing a routine prenatal care appointment after 40 weeks in three maternity units in Spain and the United Kingdom from January 2017 to December 2019. The primary outcome was adverse perinatal outcomes defined as stillbirth or neonatal death, cesarean section or instrumental delivery due to fetal distress during labor, umbilical arterial cord blood pH < 7.0, umbilical venous cord blood pH < 7.1, Apgar score at 5 min < 7, and admission to the neonatal unit. Logistic mixed models and ROC curve analyses were used to analyze the data. *Results:* A total of 3143 pregnancies were analyzed, including 537 (17.1%) with an adverse perinatal outcome. Maternal age (odds ratio (OR) 1.03, 95% confidence interval (CI) 1.01 to 1.04), body mass index (OR 1.04, 95% CI 1.03 to 1.06), racial origin (OR 2.80, 95% CI 1.90 to 4.12), parity (OR 0.36, 95% CI 0.29 to 0.45), and labor induction (OR 1.79, 95% CI 1.36 to 2.35) were significant predictors of adverse perinatal outcomes with an area under the ROC curve of 0.743 (95% CI 0.720 to 0.766). The addition of the CPR to the previous model did not improve performance. Additionally, the CPR alone achieved a detection rate of only 11.9% (95% CI 9.3 to 15) when using the 10th centile as the screen-positive cutoff. *Conclusions:* Our data on late-term unselected pregnancies suggest that the CPR is a poor predictor of adverse perinatal outcomes.

## 1. Introduction

Late-term and post-term pregnancies are known to have a higher incidence of adverse perinatal outcomes [1,2]. Unfortunately, prenatal detection of such pregnancy complications at that stage remains a challenge. Several studies have evaluated the role of fetal heart monitoring (non-stress test [NST]), ultrasound estimation of fetal weight (EFW), and Doppler assessment of the fetal umbilical artery (UA), fetal middle cerebral artery (MCA), and uterine arteries. Still, none of these methods have been proven effective in improving perinatal outcomes in late-term and post-term pregnancies [3,4,5,6]. 

In 1983, Arbelli et al. described the cerebroplacental ratio (CPR) as a measure that quantifies the cerebral centralization of fetal blood flow in response to hypoxemia, and it is calculated as the fraction between the MCA pulsatility index (PI) and the UA PI [7]. Since then, several studies have evaluated the role of the CPR as a predictor of adverse pregnancy outcomes, showing contradictory results. Although the value of fetal Doppler assessment in pregnancies complicated with fetal growth restriction is unquestionable, it is uncertain whether, in unselected populations, the CPR could help in identifying babies at increased risk of adverse outcomes [8,9,10,11,12,13].

The objective of this study was to investigate the predictive performance of the CPR in screening for adverse perinatal outcomes in an unselected late-term population. 

## 2. Materials and Methods

### 2.1. Study Design and Population

This is a retrospective cohort study performed at three fetal medicine units, including two in Spain (Hospital Clínico Universitario Virgen de la Arrixaca in Murcia and Hospital Universitario de Torrejón in Madrid) and one in the United Kingdom (Medway Maritime Hospital in Gillingham), between January 2017 and December 2019. In the participating centers, all undelivered women attend a routine ultrasound examination at 40–42 weeks’ gestation. During this visit, maternal characteristics and medical history were recorded, including maternal age, weight, height, body mass index (BMI), racial origin (White, Black, South-Asian, East-Asian, and mixed), method of conception (spontaneous or assisted), cigarette smoking (yes or no) during pregnancy, and parity (parous or nulliparous if no previous gestation with delivery after 24 weeks). Pregnancies were dated according to the measurement of the fetal crown-rump length at 11–13 weeks at the time of screening if they were naturally conceived [14] and according to the conception date if they were conceived by in vitro fertilization. For this study, we included all uncomplicated singleton pregnancies attending an appointment at 40–42 weeks’ gestation at any participating center. We excluded all cases with fetal growth disorders diagnosed in previous scans (EFW <10th or >90th centile) and those where any additional ultrasound beyond the 35–36-week scan established by the routine protocol in the participating center had been performed. By excluding these cases, we ensured that all pregnancies with any type of maternal, fetal, or pregnancy complication and any relevant risk factor were not included. Pregnancies lost to follow-up and those with planned cesarean sections were also excluded.

At the 40–42-week ultrasound, fetal weight was estimated from the measurements of the fetal head, abdominal circumference, and femur length [15]. Transabdominal Doppler assessment of umbilical artery (UA) pulsatility index (PI), middle cerebral artery (MCA) PI, and CPR was performed. Color flow mapping was used to identify the umbilical artery in a free-floating loop of the umbilical cord and the proximal segment of the MCA as it emerges from the circle of Willis in an axial section of the brain. The PI was then determined by recording at least three consecutive uniform waveforms without fetal body or respiratory movements using pulsed Doppler at an angle of insonation of less than 15º [16]. Fetal biometry and fetal–maternal Doppler ultrasound examinations were performed by certified sonographers. All pregnancies meeting the inclusion criteria during the study period were included.

### 2.2. Outcome Measures

Data on pregnancy outcomes were collected from hospital records. We recorded delivery characteristics, including onset of labor (spontaneous or induced), type of delivery (cesarean section, operative (forceps or ventouse), and non-operative vaginal delivery), and indications for them. We also recorded neonatal characteristics, such as umbilical arterial and vein cord blood pH, Apgar scores at 1 and 5 min, sex, birthweight, birthweight Z-score, and need for admission to the neonatal unit (NNU). We excluded pregnancies undergoing elective cesarean section with no previous labor.

We considered the following as adverse perinatal outcomes: stillbirth or neonatal death, cesarean section or instrumental delivery due to fetal distress, umbilical arterial cord blood pH ≤ 7, umbilical venous cord blood pH ≤ 7.1, Apgar score at 5 min < 7, and admission to the NNU.

### 2.3. Statistical Analysis

Descriptive data are expressed as median and interquartile range (IQR) and proportions (absolute and relative frequencies). Comparisons between outcome groups were performed using the Mann–Whitney U-test or Fisher test, as appropriate.

Age and BMI were centered by subtracting their mean from their raw values. CPR values were transformed to multiples of the median (MoMs) [16,17]. EFW was also transformed to its centile [18]. Since this was a predominantly White population, racial origin was grouped as “White” and “Others”. 

Two multiple logistic mixed-effects models were adjusted using adverse outcome as the dependent variable and age, BMI, racial origin, conception, smoking, parity, scan-to-birth interval, labor onset, estimated fetal weight centile, and gestational age at delivery as independent variables. Hospital of precedence was set as a random factor, and the CPR MoM was included in one model but not in the other. Odds ratios (ORs), their 95% confidence intervals (CIs), and *p*-values were computed. To test our hypothesis that the CPR significantly contributes to the prediction of adverse outcomes, the receiver-operating characteristics (ROC) curves of both models were compared visually and numerically by computing the area under the curve (AUC) and its 95% CI. Analysis was performed on a complete case basis, and the number of pregnancies included in each calculation was reported whenever necessary. The level of significance was set at 0.05. 

To assess the predictive performance of the CPR alone in the detection of adverse perinatal outcomes, we estimated the DR by setting the 10th centile as the cutoff value. We also presented the CPR MoMs for the birthweight centile.

Last, to evaluate the study dataset’s ability to detect significant differences in CPR values between outcome groups, we performed a post-hoc power analysis. 

All analyses were carried out with the statistical software R (version 4.2.2) [19]. Models were fit with the R package ‘lme4’ [20], and ROC curves were estimated using the R package ‘pROC’ [21]. 

## 3. Results

### 3.1. Study Population

After exclusions (additional scans during the second or third trimester, n = 1117; fetal growth disorders without additional scans, n = 38; lost to follow-up, n = 22; planned cesarean section, n = 72; incomplete data, n = 66), a total of 3143 consecutive pregnancies attending their 40–42-week ultrasound assessment were included in the study. The delivery occurred within one week in 98.3% of the cases. Maternal and pregnancy characteristics of the study population are provided in Table 1. Of them, 537 (17.1%) pregnancies had an adverse perinatal outcome.

Compared to uncomplicated pregnancies, those women with pregnancies ending with an adverse perinatal outcome were predominantly nulliparous, had a greater BMI, were younger and more often of non-White racial origin, and had a higher rate of induced labor.

### 3.2. CPR as a Predictor of Adverse Outcome

The multivariate logistic regression analysis identified age, BMI, non-White racial origin, multiparity, and induced labor onset as independent predictors of adverse perinatal outcomes. Of them, maternal age (OR 1.03, 95% CI 1.01 to 1.04), BMI (OR 1.04, 95% CI 1.03 to 1.06), non-White racial origin (OR 2.80, 95% CI 1.90 to 4.12), and induced labor onset (OR 1.79, 95% CI 1.36 to 2.35) increased the risk of adverse perinatal outcome, and multiparity (OR 0.36, 95% CI 0.29 to 0.45) decreased it (Table 2). CPR was not statistically significant in the multiple model (OR 1.22, 95% CI 0.89 to 1.65). The respective AUCs were identical at 0.743 (95% CI 0.720 to 0.766) for the model with CPR and the model without CPR (Appendix A).

At a 10% screen-positive rate (SPR), the detection rate (DR) of pregnancies at a higher risk of adverse outcome was 26.8% (95% CI 23.1 to 30.8), and the positive and negative likelihood ratios were 4.14 (95% CI 3.38 to 5.06) and 0.78 (95% CI 0.74 to 0.82), respectively, for the model that did not include CPR. For the model that included the CPR, the same values were 26.4% (95% CI 22.8 to 30.4), 4.03 (95% CI 3.29, 4.93), and 0.79 (95% CI 0.75 to 0.83), respectively, showing no statistically significant differences between the two models. 

At a 15% SPR, the DR was 35.9% (95% CI 31.9 to 40.2), and the positive and negative likelihood ratios were 3.38 (95% CI 2.89 to 3.96) and 0.72 (95% CI 0.67 to 0.76), respectively, for the model that did not include CPR. For the model that included the CPR, the same values were 35.6% (95% CI 31.5 to 39.8), 3.32 (95% CI 2.83 to 3.89), and 0.72 (95% CI 0.68 to 0.77), respectively, with no differences between the models. Therefore, there were no statistically significant differences in the capacity for predicting adverse perinatal outcomes after the inclusion of the CPR.

Finally, the DR for the CPR alone was 11.9% (95% CI 9.3 to 15), using the 10th centile as the screen-positive cutoff (Figure 1).

### 3.3. Post-hoc Power Analysis

The analysis indicated an 82% power to detect a mean difference of 0.08 considering a standard deviation of 0.580 and an event/no-event ratio of 0.2. With these values, we also had 90% power to detect a mean difference of 0.09 and 95% power to detect a mean difference of 0.1 CPR units, which is a minimal difference that could be clinically relevant.

## 4. Discussion

### 4.1. Main Findings

This study showed that significant predictors of adverse perinatal outcomes identified at the 40–42-week scan are maternal age, BMI, racial origin, parity, and labor onset. The addition of CPR does not improve the predictive capacity achieved by the previous factors in late-term unselected pregnancies.

The vast majority of the pregnancies ending with an adverse perinatal outcome presented a normal CPR within the previous two weeks.

### 4.2. Comparison with Previous Studies

Over the last few years, great interest has arisen in the role of CPR in identifying pregnancies at high risk of adverse pregnancy outcomes with appropriately grown fetuses based on the assumption that late-onset placental insufficiency may not be associated with growth restriction but may still increase the risk of adverse perinatal outcomes [22]. Since a low CPR indicates a redistribution of the cardiac output to the brain, it has been hypothesized that this could be used as a proxy for hypoxemia and, therefore, poorer perinatal outcome. 

Recent evidence has shown that CPR per se may be of value in detecting pregnancies with placental insufficiency and fetal hypoxemia, as it is able to predict adverse perinatal events in fetuses with adequate weight. In a meta-analysis including 47 studies and more than 66,000 patients, CPR was a significant predictor of operative delivery for non-reassuring fetal status (RR 2.52, 95% CI 2.10 to 3.02), umbilical cord blood pH < 7 (RR 2.19, 95% CI 1.01 to 4.75), and low Apgar score (RR 2.05, 95% CI 1.39 to 3.03) [10]. A more recent systematic review by Elmes et al. described CPR as the best predictor of fetal demise (OR 4.02, 95% CI 1.32 to 14.4) and composite adverse outcomes (OR 6.28, 95% CI 2.67 to 10.15). Still, it did not perform as well in predicting operative vaginal delivery or admission to the NNU (OR 3.13, 95% CI 1.31 to 9.73 and OR 3.78, 95% CI 1.48 to 9.47, respectively) [23]. Although there seems to be an association between low CPR values and both stillbirth and adverse perinatal outcomes, it still appears to be a weak predictor, and active management based solely on the CPR is currently not recommended [24]. In our study, we were not able to analyze the predictive capacity of the CPR for fetal demise because we did not have a sufficient number of stillbirth and neonatal death cases. Regarding the surrogate markers of fetal hypoxia (low pH or Apgar, cesarean for fetal distress, and NNU admission), the CPR did not perform as well as described in the systematic review of Elmes et al., and its inclusion did not improve the prediction provided by maternal and pregnancy characteristics.

Some studies found the CPR to be a good predictor of adverse perinatal outcomes alone or in combination with EFW [22,25]. Flatley et al. analyzed 2425 gestations and reported a DR of 23.3% for the CPR alone, which increased to 36.7% when combined with EFW below the 10th centile, improving throughout pregnancy [26]. Other smaller studies have analyzed pregnant women at greater than 40 weeks of gestation, showing contradictory results. Ropacka et al. described that, in 148 women, the CPR at 40–42 weeks had DRs of 74.1% for predicting intrapartum abnormal fetal heart rate and 87.8% for predicting adverse neonatal outcomes [27]. Morales-Roselló et al. analyzed 569 pregnancies between 36 and 40+6 weeks and reported a DR of 30% for adverse perinatal outcomes [9]. However, the pregnancies assessed in these studies did not correspond to a routine population but a preselected one, and Morales-Roselló et al. also included third-trimester gestations (mean of 38 weeks at ultrasound). 

On the other hand, several studies have not demonstrated the association between low CPR and adverse perinatal outcomes in low-risk pregnancies (late-term pregnancies or during the third trimester). Ortiz et al. measured the CPR in 314 late-term pregnancies (41–41+6) before induction of labor and described a low predictive value for operative delivery due to intrapartum fetal compromise and adverse perinatal outcome (DRs of 26% and 19% at 13% and 16% false positive rates, respectively) [28]. Lebovitz et al. included 120 gestations exceeding 40 weeks and showed that the CPR was not associated with adverse perinatal outcomes in low-risk pregnancies or in the subgroup of small for gestational age fetuses [11]. Villalain et al. in 2021 described the CPR as a poor predictor of adverse perinatal outcomes regardless of fetal weight, with AUCs of 0.44 (95% CI 0.39 to 0.51) adequate for gestational age fetuses and 0.56 (95% CI 0.49 to 0.61) for late fetal growth restriction [29]. 

Consistent with our findings, the largest published studies have demonstrated poor predictive performance of the CPR for adverse perinatal outcomes [30,31]. Akolekar et al. studied the CPR in over 47,000 gestations from 35 to 37 weeks and found limited predictive accuracy, with a modest improvement if the delivery occurred within the following two weeks after assessment [8]. However, we did not find any association, even if 98.3% of our patients delivered within one week. A 2020 meta-analysis that included 18,700 patients concluded that the CPR was associated with adverse perinatal outcomes, but it did not perform better than use of the umbilical artery alone [13]. 

### 4.3. Strengths and Limitations

To the best of our knowledge, this is the largest study assessing the value of the CPR in unselected pregnant women attending a routine ultrasound examination within the last two weeks of their pregnancy. By focusing on the gestational age range of 40–42 weeks, we were able to investigate the CPR near delivery, ensuring that the Doppler evaluation truly reflects placental function at the time of admission. 

The main limitation of this study is related to its observational and retrospective nature, which limits the inclusion of cases to those with all variables recorded. Additionally, the limited number of cases of each adverse outcome individually (stillbirth/neonatal death, low cord pH or Apgar score, fetal distress, or admission to the NNU) prevented subgroup analysis. Finally, in the assessment of the CPR as a predictor of such perinatal outcomes, it is essential to acknowledge that all of them may be influenced by the various external factors occurring during labor and delivery. This could result in an underestimation of the predictive capacity of the CPR as a marker of placental dysfunction.

### 4.4. Clinical Implications

Considering that the CPR has not been demonstrated to improve the detection of cases ending in adverse perinatal outcomes in unselected populations, its routine indiscriminate use for clinical management of pregnancies in the absence of fetal growth restriction is likely to increase obstetric intervention without any improvement in clinical outcomes. However, further studies should evaluate intrapartum variables together with the CPR to predict adverse outcomes.

On the other hand, the CPR has proven its value in the management of small for gestational age fetuses. Therefore, it may also benefit other high-risk women, such as those presenting with decreased fetal movements or light vaginal bleeding or those undergoing labor induction for different reasons. Future research is needed to evaluate if incorporating the CPR measurement in those cases would improve fetal assessment and facilitate the early detection of potential complications.

## 5. Conclusions

Maternal age, BMI, racial origin, parity, and labor induction are significant predictors of adverse perinatal outcomes, defined as perinatal death, cesarean section for fetal distress, low blood cord pH, low 5 min Apgar score, or NNU admission, in low-risk pregnancies assessed at term. However, the value of the CPR for this purpose is limited.

## Figures and Tables

**Figure 1 medicina-59-01670-f001:**
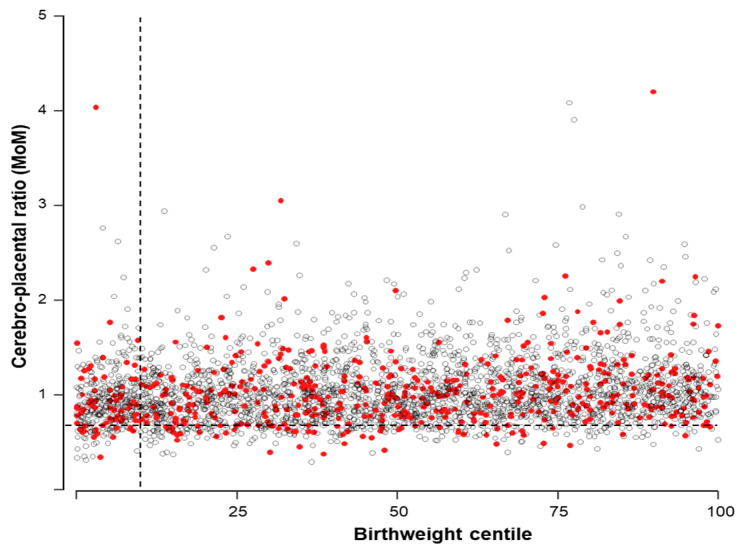
Distribution of the 3143 pregnancies according to birthweight and cerebroplacental ratio multiples of the median (MoMs). White dots represent uneventful cases, and red dots are those ending with an adverse pregnancy outcome. The interrupted horizontal line represents the 10th CPR MoM, and the interrupted vertical line represents the 10th birthweight centile.

**Table 1 medicina-59-01670-t001:** Antenatal and natal characteristics of the study population.

	No Adverse Event	Adverse Event	*p*-Value
	(N = 2606)	(N = 537)	
Age (years)	31.2 [27.0, 35.0]	30.5 [26.4, 34.6]	0.0506
Body Mass Index (km/m^2^)	26.7 [23.2, 31.3]	29.8 [26.1, 35.0]	**<0.001**
Racial origin			
White	2523 (96.8%)	482 (89.8%)	**<0.001**
Other	83 (3.2%)	55 (10.2%)	
Cigarette smoker			
No	2325 (89.2%)	485 (90.3%)	0.489
Smoker	281 (10.8%)	52 (9.7%)	
Conception			
Spontaneous	2485 (95.4%)	503 (93.7%)	0.101
No spontaneous	121 (4.6%)	34 (6.3%)	
Parity			
Multip	1337 (51.3%)	167 (31.1%)	**<0.001**
Nullip	1269 (48.7%)	370 (68.9%)	
Scan-to-birth interval (days)	1.00 [0.200, 3.00]	0.500 [0.100, 1.00]	**<0.001**
Estimated fetal weight centile	48.9 [19.4, 77.2]	51.1 [20.8, 80.1]	0.407
Birthweight centile	48.2 [23.0, 72.2]	48.0 [22.4, 73.9]	0.895
CPR (MoM)	0.977 [0.809, 1.20]	0.978 [0.784, 1.17]	0.138
Gestational age at delivery (weeks)	41.1 [40.6, 41.5]	41.2 [40.3, 41.6]	0.589
Labor onset			
Spontaneous	893 (34.3%)	79 (14.7%)	**<0.001**
Induced	1713 (65.7%)	458 (85.3%)	
Mode of delivery			
Vaginal	2222 (85.3%)	141 (26.3%)	**<0.001**
CS for FD	0 (0%)	336 (62.6%)	
CS for other	384 (14.7%)	60 (11.2%)	
Venous pH ≤ 7.1			
No	2606 (100%)	517 (96.3%)	**<0.001**
Venous pH ≤ 7.1	0 (0%)	20 (3.7%)	
Arterial pH ≤ 7			
No	2606 (100%)	515 (95.9%)	**<0.001**
Arterial pH ≤ 7	0 (0%)	22 (4.1%)	
Stillbirth or admission to the neonatal unit			
No	2606 (100%)	320 (59.6%)	**<0.001**
Yes	0 (0%)	217 (40.4%)	
5 min Apgar			
≥7	2606 (100%)	513 (95.5%)	**<0.001**
<7	0 (0%)	24 (4.5%)	

CPR, cerebroplacental ratio; MoM, multiple of the median. Highlighted in bold *p* < 0.05.

**Table 2 medicina-59-01670-t002:** Results from logistic mixed models.

	Univariate			Multiple (without CPR)			Multiple (with CPR)		
Predictors	Odds Ratios	CI	*p*-Value	Odds Ratios	CI	*p*-Value	Odds Ratios	CI	*p*-Value
Intercept				0.20	0.12 to 0.35	**<0.001**	0.17	0.09 to 0.32	**<0.001**
Age (years)	1.01	0.99 to 1.02	0.493	1.03	1.01 to 1.04	**0.009**	1.03	1.01 to 1.05	**0.007**
Body mass index (km/m^2^)	1.04	1.02 to 1.05	**<0.001**	1.04	1.03 to 1.06	**<0.001**	1.04	1.03 to 1.06	**<0.001**
Racial origin									
White	Reference			Reference			Reference		
Other	2.31	1.61 to 3.33	**<0.001**	2.80	1.90 to 4.12	**<0.001**	2.81	1.91 to 4.14	**<0.001**
Cigarette smoker									
No	Reference			Reference			Reference		
Smoker	0.85	0.62 to 1.17	0.319	1.05	0.75 to 1.47	0.777	1.06	0.76 to 1.48	0.743
Conception									
Natural	Reference			Reference			Reference		
Other	1.38	0.92 to 2.07	0.117	1.01	0.65 to 1.57	0.964	1.01	0.65 to 1.57	0.971
Parity									
Nulliparous	Reference			Reference			Reference		
Parous	0.39	0.31 to 0.47	**<0.001**	0.36	0.29 to 0.45	**<0.001**	0.36	0.29 to 0.45	**<0.001**
Scan-to-birth interval (days)	1.00	0.92 to 1.07	0.925	0.93	0.85 to 1.00	0.060	0.93	0.85 to 1.00	0.061
Labor onset									
Spontaneous	Reference			Reference			Reference		
Induced	2.04	1.57 to 2.66	**<0.001**	1.79	1.36 to 2.35	**<0.001**	1.79	1.36 to 2.35	**<0.001**
Gestational age at delivery (weeks)	1.10	0.97 to 1.25	0.139	1.08	0.95 to 1.24	0.242	1.08	0.94 to 1.24	0.257
Estimated fetal weight centile	1.00	1.00 to 1.00	0.394	0.95	0.88 to 1.03	0.238	0.94	0.87 to 1.03	0.176
CPR (MoM)	1.06	0.78 to 1.42	0.719				1.22	0.89 to 1.65	0.212

The dependent variable is adverse outcome. Random effects are the hospitals. CI, confidence interval; CPR, cerebroplacental ratio; MoM, multiple of the median. Highlighted in bold *p* < 0.05.

## Data Availability

The data presented in this study are available on request from the corresponding author. The data are not publicly available due to local policies on data protection and privacy.

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
