# Peer review of "Efficiency of the Cerebroplacental Ratio in Identifying High-Risk Late-Term Pregnancies"

_medicina, 2023, doi:10.3390/medicina59091670_

Round 1

Reviewer 1 Report

The objective of this study was to investigate the predictive performance of the CPR in screening for adverse perinatal outcomes in an unselected late- and post-term populations. However, there are some queries about the methodology of the study. How was the sample size calculated? It's necessary to have an adequate sample size before conclusion of the insignificant findings. Please reconsider this issue as this is the most important part of the study and affect the results of the study which is different from previous studies.

Author Response

Thank you for your comment. We have included a post-hoc sample power analysis in the Methods and Results.

To evaluate the study dataset's ability to detect significant differences in CPR values between outcome groups, we performed a post-hoc power analysis.

The analysis indicated an 82% power to detect a mean difference of 0.08, considering a standard deviation of 0.580 and an event/no-event ratio of 0.2. With these values, we also had 90% power to detect a mean difference of 0.09 and 95% power to detect a mean difference of 0.1 CPR units.

Reviewer 2 Report

I must confess that I was surprised by the result of your study.

Until now, all the studies I read were very favorable for CPR's importance in evaluating pregnancies at risk. I checked your statistical data and indeed it seems you are right. CPR is not that important. I notice that your study is the largest published until now on this subject, so your results are more precise than others.

Congratulations!

English is fine

Author Response

Thank you very much for taking the time to review our article. There is indeed a good place for the CPR in the follow-up of small for gestational age babies but the studies looking at its contribution in routine populations showed contradictory results. We believe that this large study in late- and post-term pregnancies gives an almost final answer to this question.

Reviewer 3 Report

I have reviewed the manuscript titled, "Efficiency of the Cerebro-Placental Ratio in Identifying High-Risk Late- and Post-Term Pregnancies."

This study investigates the effectiveness of the cerebro-placental ratio (CPR) in forecasting adverse perinatal outcomes for pregnancies between 40-42 weeks of gestation. While several predictors like maternal age, body mass index, racial origin, parity, and labor induction were identified, incorporating CPR with these predictors did not enhance the predictive model's accuracy. When CPR was employed individually, it detected adverse perinatal outcomes at a rate of just 11.9%.

The structure of the study is clear, segmented appropriately into the background, objective, methodology, results, and conclusion, facilitating an easy grasp of the research's intent and outcomes. Nonetheless, there are areas that require attention:

1- Introduction:

-       The background review of extant literature needs a more comprehensive scan. I recommend - incorporating broader citations and considering the current standards for US biomarkers as indicated by the following: [doi: 10.1002/uog.26092], [doi: 10.1016/j.ajog.2018.01.011], and [doi: 10.1016/j.ajog.2015.10.931]. This will reinforce the argument for refining CPR standards interpretation during late gestational periods.

-       The phrase "unselected population" warrants clarification. Does it refer to a cohort that wasn't pre-filtered based on certain risk parameters?

2- Materials and Methods:

-       Clearly outline both the inclusion and exclusion criteria. Incorporating a PRISMA flowchart can demystify the process, offering readers clarity on the criteria essential for interpreting results.

3- Results:

-       Parameters that weren't significant in the univariate analysis should be excluded from the multivariate analysis. Please elucidate your rationale behind this decision. A post hoc sample size analysis might be beneficial in this context.

-       Refine Figure 1 for clarity, as the characters are currently diminutive, impeding comprehension.

-       Consider presenting the model's ROC curve, even as supplementary material, to aid in AUC interpretation. Also, both +LRR and -LRR should be included for the models.

4- Discussion:

-       Incorporate a section that delves into the biological reasoning behind the findings. Elaborate on why cerebroplacental blood flow redistribution serves as an indirect indicator of compromised fetal growth and intrauterine hypoxia, as substantiated by [doi: 10.1007/s40291-022-00611-4] and [doi: 10.1097/GCO.0000000000000490].

5- Ethical Approval:

-       Lines 263-264: The IRB approval date and protocol name reported for one of the participating centers appear to be placeholders. Kindly update this information.

Moderate Editing of English language and style is required.

Author Response

Responses to reviewer 3

1- Introduction:

-       The background review of extant literature needs a more comprehensive scan. I recommend - incorporating broader citations and considering the current standards for US biomarkers as indicated by the following: [doi: 10.1002/uog.26092], [doi: 10.1016/j.ajog.2018.01.011], and [doi: 10.1016/j.ajog.2015.10.931]. This will reinforce the argument for refining CPR standards interpretation during late gestational periods.

Thank you for your comment and the provided references. We have reviewed the referred articles which deal with uterine artery pulsatility index (reference ranges) and the value of EFW (reference ranges and comparison of INTERGROWTH-21 vs customized charts). In this study we are not assessing the value of any of those variables, and we are focusing on routine population, pregnancies not classified with any complication (including fetal growth) before attending a routine 40-42 weeks scan. We believe that focusing on our main objective will keep the manuscript simpler in this occasion.

-       The phrase "unselected population" warrants clarification. Does it refer to a cohort that wasn't pre-filtered based on certain risk parameters?

Thank you, we have clarified that in the Methods section, “Study design and population”.

For this study, we included all uncomplicated singleton pregnancies attending a 40-42 weeks’ appointment at any of the participating centers. We excluded all cases diagnosed with fetal growth disorders diagnosed in previous scans (EFW <10th or >90th centile) and those where any additional ultrasound beyond the 35-36 weeks scan established by the routine protocol in the participating center had been performed. By excluding these cases we ensured that all pregnancies with any type of maternal, fetal or pregnancy complication as well as any relevant risk factor were not included.

2- Materials and Methods:

-       Clearly outline both the inclusion and exclusion criteria. Incorporating a PRISMA flowchart can demystify the process, offering readers clarity on the criteria essential for interpreting results.

Thank you for the appreciation. We have expanded the Methods as shown in the previous question and reported the several reasons for exclusion in the first sentence of the Results.

After exclusions (additional scans during the second or third trimester, n = 1117, fetal growth disorders without additional scans, n = 38, lost-to-follow-up, n = 22, planned cesarean section, n = 72, incomplete data, n = 66),

3- Results:

-       Parameters that weren't significant in the univariate analysis should be excluded from the multivariate analysis. Please elucidate your rationale behind this decision. A post hoc sample size analysis might be beneficial in this context.

Thank you for your comment. We are not completely sure we have understood it. In agreement with current advice for adjusting multivariate analysis, we did not exclude the variables that were not significant in the multivariate model. In this paper, under the section “Excluding covariates that are non-significant in the (final) model” there is a more detailed explanation. https://academic.oup.com/ejcts/article/55/2/179/5265263

We have included a post-hoc sample power analysis in the Methods and Results.

To evaluate the study dataset's ability to detect significant differences in CPR values between outcome groups, we performed a post-hoc power analysis.

The analysis indicated an 82% power to detect a mean difference of 0.08, considering a standard deviation of 0.580 and an event/no-event ratio of 0.2. With these values, we also had 90% power to detect a mean difference of 0.09 and 95% power to detect a mean difference of 0.1 CPR units.

-       Refine Figure 1 for clarity, as the characters are currently diminutive, impeding comprehension.

Thank you, we have enlarged the image to make it clearer.

-       Consider presenting the model's ROC curve, even as supplementary material, to aid in AUC interpretation. Also, both +LRR and -LRR should be included for the models.

Thank you. We have included Figure S1 and LRs in the Results.

4- Discussion:

-       Incorporate a section that delves into the biological reasoning behind the findings. Elaborate on why cerebroplacental blood flow redistribution serves as an indirect indicator of compromised fetal growth and intrauterine hypoxia, as substantiated by [doi: 10.1007/s40291-022-00611-4] and [doi: 10.1097/GCO.0000000000000490].

Thank you. We have added two new sentences to delve into this issue.

Since a low CPR indicates redistribution of the cardiac output to the brain, it has been hypothesized that this could be used as a proxy for hypoxemia and, therefore, poorer perinatal outcome.

Although there seems to be an association between low CPR values and both, stillbirth, and adverse perinatal outcome, it still seems a weak predictor and active management based solely on the CPR is currently not recommended [24].

5- Ethical Approval:

-       Lines 263-264: The IRB approval date and protocol name reported for one of the participating centers appear to be placeholders. Kindly update this information.

Thank you for noticing. We have completed that information.

Comments on the Quality of English Language: Moderate Editing of English language and style is required.

A second native speaker has reviewed the paper and suggested a few modifications throughout the manuscript.

Round 2

Reviewer 1 Report

The authors described postdoc power analysis for detection of the CPR difference between two groups. However, in this study, the CPR difference was 0.001 MoM. The power of this sample size was therefore too low to detect the CPR difference.

As the predictive performance of CPR is the main objective of this study, the sample size to detect the CPR difference between two groups should be adequate. Please reconsider this important issue.

Author Response

Thank you for your comment. Our sample has a power of 95% to detect a mean difference of 0.1 CPR MoMs, which seems like a very conservative threshold to be considered clinically relevant. Differences of 0.001 are most likely related to biological variability.

The analysis indicated an 82% power to detect a mean difference of 0.08, considering a standard deviation of 0.580 and an event/no-event ratio of 0.2. With these values, we also had 90% power to detect a mean difference of 0.09 and 95% power to detect a mean difference of 0.1 CPR units, which is a minimal difference that could be clinically relevant.